# Non-Antibiotic Antimony-Based Antimicrobials

**DOI:** 10.3390/molecules27217171

**Published:** 2022-10-23

**Authors:** Nikolay Gerasimchuk, Kevin Pinks, Tarosha Salpadoru, Kaitlyn Cotton, Olga Michka, Marianna A. Patrauchan, Karen L. Wozniak

**Affiliations:** 1Department of Chemistry and Biochemistry, Temple Hall 456, Missouri State University, Springfield, MO 65897, USA; 2Department of Microbiology and Molecular Genetics, Oklahoma State University, Stillwater, OK 74078, USA

**Keywords:** cyanoximes, organometallic compounds of antimony(V), crystal structures, antimicrobial activity, antibacterial activity, antifungal activity

## Abstract

A series of the eight novel organoantimony(V) cyanoximates of Sb(C_6_H_5_)_4_L composition was synthesized using the high-yield heterogeneous metathesis reaction between solid AgL (or TlL) and Sb(C_6_H_5_)_4_Br in CH_3_CN at room temperature. Cyanoximes L were specially selected from a large group of 48 known compounds of this subclass of oximes on the basis of their water solubility and history of prior biological activity. The synthesized compounds are well soluble in organic solvents and were studied using a variety of conventional spectroscopic and physical methods. The crystal structures of all reported organometallic compounds were determined and revealed the formation of the distorted trigonal bipyramidal environment of the Sb atom and monodentate axial binding of acido-ligands via the O atom of the oxime group. The compounds are thermally stable in the solid state and in solution molecular compounds. For the first time, this specially designed series of organoantimony(V) compounds is investigated as potential non-antibiotic antimicrobial agents against three bacterial and two fungal human pathogens known for their increasing antimicrobial resistance. Bacterial pathogens included Gram-negative Escherichia coli and Pseudomonas aeruginosa, and Gram-positive Staphylococcus aureus. Fungal pathogens included Cryptococcus neoformans and Candida albicans. The cyanoximates alone showed no antimicrobial impact, and the incorporation of the SbPh_4_ group enabled the antimicrobial effect. Overall, the new antimony compounds showed a strong potential as both broad- and narrow-spectrum antimicrobials against selected bacterial and fundal pathogens and provide insights for further synthetic modifications of the compounds to increase their activities.

## 1. Introduction

Microbial infections are associated with considerable morbidity and costs. The continuous spread of antimicrobial resistance (AMR) among bacterial and fungal pathogens imperils the usefulness of antibiotics and antifungals, which previously revolutionized and enabled medical interventions. The increasing AMR is manifesting as multidrug resistance (MDR), leading to a global crisis that, if not addressed, will soon have a huge adversarial impact on human race. The main causes of the rapid increase in AMR include an overuse of antimicrobials, inappropriate prescribing, and the inability of patients to follow the prescribed drug regimen. Extensive use of antibiotics in agricultural settings and the wide availability of some antibiotics are also important factors. This situation led to the appearance of antibiotic resistant bacterial pathogens, exemplified by methicillin-resistant *Staphylococcus aureus* (MRSA), and vancomycin-resistant enterococci (VRE), which are becoming increasingly difficult to treat with the current range of available antibiotics. Fungal pathogens *Cryptococcus neoformans* and *Candida albicans* are also becoming increasingly resistant to antifungal drugs, and different *Candida* species with natural drug resistance, such as *Candida auris*, are evolving in hospital settings and becoming major pathogens. The MDR phenomenon is one of the greatest global public health challenges that humanity is currently facing. According to the Centers for Disease Control and Prevention (CDC) report on antimicrobial resistance conducted in 2019, annually, there are more than 2.8 million antibiotic-resistant bacterial infections, from which more than 35,000 people die in the US. Although fungal drug-resistant strains are more difficult to track, current estimates indicate that fungal infections lead to more than 1.5 million deaths annually worldwide. Additionally, the economic impact due to treating these infections, as well as the loss of productivity, is astronomical—about 4.6 billion USD annually for bacterial infections and about 7.2 billion USD annually for fungal infections.

With modern medicine’s reliance on antimicrobials in treating diseases such as pneumonia, tuberculosis, sexually transmitted diseases, and bloodstream infections, the problem with AMR is that, soon, no current treatments will be effective. Therefore, the development of new antimicrobial agents is imperative. The critically important progress in this field requires the development of new chemical compounds that would inhibit bacterial and/or fungal growth while not being toxic to human tissues. Pure organic compounds, which classic antibiotics are, cannot meet these criteria, as they can be metabolized by microorganisms or lead to the development of resistance. In contrast, inorganic salts and Werner-type complexes may offer an alternative with the required properties.

The successful application of metal complexes and organometallic compounds in the treatment of numerous human diseases is a vigorously expanding area in both biomedical and bioinorganic chemistry research. Remarkably, the group XV elements in the periodic table—pnictogens—contain all biologically active elements. Nitrogen (N) and phosphorus (P) are crucial for life on the planet, while arsenic (As), antimony (Sb), and bismuth (Bi) possess very useful properties for various biomedical applications. Paul Ehrlich, widely known as the father of antimicrobial chemotherapy and bioinorganic chemistry, found the “magic bullet” (compound 606) capable of treating the highly infectious bacterial pathogen *Treponema pallidum* without hurting the host. This compound is a heterocyclic arsenic-based small molecule (C_12_H_13_As_2_C_l_N_2_O_2_) that was used to successfully treat syphilis and was marketed as Salvarsan (“lifesaving”) [1]. Despite the well-known toxicity of arsenic, this drug saved >560,000 lives within 50 years [2]. Bismuth, contrary to arsenic, is not toxic and is widely used as subsalicylate (C_7_H_5_BiO_4_) in the well-known drug Pepto-Bismol [3]. In turn, antimony has been used since early Egyptian’s civilization [4]. For example, NaSbO_3_ was commonly used as an emetic compound until the late 1700s [5]. The other successful application of Sb is the treatment of leishmaniosis, caused by a protozoan parasite *Leishmania* transmitted through the bite of infected sandflies in South and Central America, Bangladesh, southern Europe, and North Africa. The active compounds against the disease were found to be several Sb(V) carbohydrates, such as sodium antimony gluconate (Pentostam) and meglumine antimonate (Glucantime). These have been in use for more than six decades to treat leishmaniosis [6]. In all these active Werner-type complexes, there is an [SbO6] environment. Furthermore, several organoantimony compounds have been studied and shown to possess antimicrobial, antifungal, and antitumor activities [7,8,9,10,11,12,13]. In such case, the chemical (toward hydrolysis) and thermodynamic stability of the Sb–C bond seems to be a favorable factor for biological activity and lessening of side-effects contrary to the abovementioned Sb-carbohydrates.

Cyanoximes (Figure 1) represent a new subclass of small organic molecules—oximes, the chemistry and applications of which have been intensely developed during the last two decades [14,15,16,17]. Cyanoximes and their metal salts and Werner-type complexes demonstrate a large spectrum of biological activity: pesticide antidotes [18,19], antimicrobial [20,21], and growth regulation in plants [22,23]. Recently, it was discovered that Sb(V) cyanoximates are thermally and chemically stable in both solid state and solutions [24].

In addition, no intrinsic in vitro cytotoxicity was detected for free cyanoximes, organic ligands used to synthesize organoantimony(V) compounds [25,26,27]. Thus, it is an attractive idea to explore the potential synergistic effect of both by combining the known biological activity of cyanoximes with the established medical use of antimony(V) compounds. The aim of this study was to develop new, non-antibiotic antimicrobial compounds that efficiently inhibit pathogenic bacteria and fungi. On the basis of our previous biomedical investigations of 48 currently known cyanoximes [14], we selected eight cyanoximes shown in Figure 2 for the preparation, characterization, and subsequent antimicrobial studies of new tetraphenyl-antimony(V) cyanoximates. The choice of these acido-ligands was founded primarily on their water solubility and biological activity. Furthermore, using the series of novel organoantimony compounds, we initiated the investigation of the structure (biological)–activity relationship (SAR), focusing on the effect of O vs. S atoms in the structure, bulkiness of the cyanoxime, or effect (if any) of cyclic groups vs. alkyl groups present in the ligand. In this paper, we present the preparation, characterization, and biological activity studies of the first generation (designated as G1) of new tetraphenyl-antimony(V) cyanoximates SbPh_4_L.

## 2. Results and Discussion


**
Chemical Aspect
**


The starting compound SbPh_4_Br is only one commercially available source of pentavalent antimony with four organic groups attached. In this project, we used this compound for the preparation of the first series (G1) of organoantimony(V) cyanoximates for systematic antimicrobial studies. In the past, we found that the metathesis reaction of beforehand prepared Ag(I) of cyanoxime salts with other organoelemental compounds of Te(IV) [28], Sn(IV) [26,29], and Sb(V) [24] was a very successful approach for the high-yield synthesis of desired elementorganic cyanoximates (Figure 1). Quantitative preparation of silver(I) cyanoximates has been developed in our group [30,31,32,33] (Figure 2A). In this work, we expanded the utility of the metathesis reaction in dry organic solvents with Tl(I) cyanoximates (Figure 1) and applied it to synthesize a series of Sb(V) compounds. These compounds were prepared practically quantitatively in warm aqueous solutions using Tl_2_CO_3_ as the source for the heavy-metal cation [16,34,35] (Figure 2B). Thus, we showed that the use of Tl(I) salts is equally useful and convenient when desired Ag-salts for other cyanoximates are either not obtainable or difficult to handle because of their high light sensitivity and thermal instability. Moreover, thallium(I) is not an oxidizer contrary to silver(I) salts, while Tl(I) compounds are not light sensitive and thermally stable [36], yet quantitatively form halogenides insoluble in water and common organic solvents. These factors make work with Tl(I) derivatives attractive; however, all necessary precautions should be taken due to their toxicity as explained in the *Safety Note* in Section 3. 

Crystal structures were determined for all eight obtained organoantimony(V) compounds, but only six are presented in this work (Figure 3). Two structures with thioamide-cyanoximes TCO^−^ and TDCO^−^ showed interesting peculiarities (polymorphism) and will be published in a more specialized in crystallography journal. However, both structures are only briefly presented in the Appendix A. In light of the description of this new family of Sb-based organometallic compounds, it is necessary to mention that previous biomedical research with this element focused only on compounds with traditional donor–acceptor or covalent Sb–O bonds. Thus, Sb–C bonding in organoantimony compounds is different from classic complexes and is significantly stronger (Figure 4). Organometallic compounds have great advantages in such a situation. This makes these compounds stable with respect to hydrolysis reactions in aqueous media. In all eight obtained and studied tetraphenyl-antimony(V) cyanoximates, the anion is bound to the central atom in monodentate fashion via the oxygen atom of the *oxime* group (Figure 3) as can be judged from longer N–O bonds in comparison with the C–N bond in the cyanoxime molecule.

The best description of the geometry of molecular geometries of all studied SbPh_4_L (L = cyanoxime from Figure 2) is distorted trigonal bipyramid (Figure 3). The central Sb-atom is outside of the equatorial plane in all structures and shifted toward the axial phenyl group. Thus, there is a certain “off-plane” distance in the [SbC3] equatorial environment. The geometries and the environment of Sb-centers in all compounds reported here are presented in detail pages of Appendix A.

Further analysis of the crystal structures revealed a correlation between structures and pK values of cyanoximes coordinated to the metalloid atom (Figure 5 and Figure 6). Thus, an inverse from the expected trend was observed; a more acidic cyanoxime bound to the metalloid led to a longer Sb–O bond length (Figure 5). Similarly, an inverse from expected trend was observed during analysis of the acidity of the cyanoxime and deviation from planarity of the [SbC3] base. Here, a more acidic cyanoxime bound to the metalloid led to the Sb-atom being more pushed inside the trigonal pyramidal cage toward the axial phenyl group (Figure 6).

To continue characterization of the new family of new compounds, we investigated the thermal stability of synthesized SbPh_4_L both in solutions and in a solid state. This is important for two reasons: (1) application of these new antimicrobial compounds may include using them as additives in surface coating and making wound dressing materials suitable for thermal sterilization; (2) the stability and chemical integrity in solutions at variable temperatures are important for their shelf lifetime and usability. We determined that compounds are soluble in a variety of solvents and form molecular solutions with no electrical conductivity.

The results evidenced a satisfactory stability of pure solid samples of all eight SbPh_4_L at ~150 °C, which makes them suitable for applications that require heating or heat sterilization. The TG/DSC traces for one compound showing phase transition (melting) are presented in Appendix A, accompanied by the summary of thermal stability of studied compounds. Similarly, synthesized compounds were stable in high-boiling-point solvents such as DMSO and propionitrile. More specifically, the method of variable-temperature UV/visible spectroscopy was used in this study. Thus, all obtained organoanimtony(V) cyanoximates represent colorless compounds (except for SbPh_4_(TCO) and SbPh_4_(TDCO), which were yellow because of the color of the thio–cyanoxime ligands TCO^−^ and TDCO^−^ (Appendix A)). However, cyanoxime anions ACO^−^, ECO^−^, and MCO^−^, and isomeric pyridyl–cyanoximes 2PCO^−^, 3PCO^−^, and 4PCO^−^ in chosen solvents appeared rose-pink due to the *n*→π* transition in the visible region of the spectrum [37], which is sensitive to the nature of the solvent. Therefore, the appearance of color in colorless solutions of our SbPh_4_L would be indicative of their dissociation and liberation of the colored anion in solution (Figure 1). 

Furthermore, we performed recording of the UV/visible spectra of all colorless organoantimony(V) cyanoximates in selected solvents from room temperature to 80 °C in 10 °C intervals. The results indicate the absence of appreciable ionization of cyanoximes in solutions even at elevated temperatures (Figure 7). This assures that the observed antimicrobial effects are associated with the molecular structure of the parent compound SbPh_4_L (L = studied cyanoximes anion) and not to separate SbPh_4_^+^ and L^−^ ions. 

All synthesized tetraphenyl-antimony(V) compounds were characterized by ^13^C{^1^H}-NMR spectroscopy. First of all, we should note that, despite considerable concentrations of complexes used during the NMR analysis, we had difficulties in obtaining high-resolution and good-quality spectra even after ~12 h of accumulation. We attribute this unexpected difficulty to the effect of quadrupole nuclei of ^121^Sb (57.3% abundance; *I* = +5/2) and ^123^Sb (42.7 % abundance; *I* = +7/2) isotopes that most likely adversely affected the relaxation times of carbon nuclei. 

Since it was such an unusual finding, we validated the observations by independently recording ^13^C{^1^H}-NMR spectra at different locations (Courtesy of Dr. Sergiy Tyukhtenko, Center for Drug Discovery, Northeastern University), but unfortunately with the same result. In comparison to a typical ^13^C-NMR spectrum displayed in Appendix A, we observed an interesting correlation between position of the *ipso*-carbon atom of the phenyl-groups in SbPh_4_L (L = cyanoximes from Figure 2) and pK_a_ values of the coordinated to Sb-center cyanoximes as shown in Appendix A. There were two clearly different trends: one for isomeric heterocyclic cyanoximes and another for other carboxylates and amides. No immediate rationale for the observed behavior is available at the moment.


**
Biological Aspect
**


**Antibacterial potential of new tetraphenyl-antimony(V) cyanoximates.** It should be especially mentioned that the new cyanoximates synthesized and characterized in this work represent organometallic compounds with relatively strong Sb–C covalent bonds. They do not undergo hydrolysis in the cell contrary to previously known above-cited Werner-type complexes containing labile Sb–O bonds. The presence of lipophilic organometallic fragment provides different and favorable kinetics for the intake of tetraphenyl-antimony(V) cyanoximates into cells, which helps in the delivery of biologically active cyanoxime. Furthermore, these types of small-molecule compounds were never previously exposed to pathogenic microorganisms, which is an advantage of using these new antimicrobial compounds until resistance to them is potentially developed decades later. 

To determine the antibacterial potential of the novel antimony compounds, we selected three human pathogens: Gram-negative *E. coli* and *P. aeruginosa*, and Gram-positive *S. aureus*. The Shiga toxin-producing *E. coli* strain S17 (O113:H4) was isolated from a chick liver with septicemia [38], the lab-adapted *P. aeruginosa* strain PAO1 was originally isolated from a burn-wound infection [39], and the methicillin-resistant *S. aureus* strain NRS70 was isolated from a respiratory infection [40]. These pathogens represent different infection profiles. Both *P. aeruginosa* and *S. aureus* are becoming increasingly resistant to currently available antibiotics [41,42] and were recognized by the CDC as critically important for discovering novel therapeutics. All three strains were subjected to disc-diffusion assay (Table 1, Appendix A). 

Among the antimicrobial compounds tested, SbPh_4_(ACO) and SbPh_4_(ECO) showed activity against all three bacterial pathogens. SbPh_4_(MCO) was effective against *P. aeruginosa* and *S. aureus*, and SbPh_4_(TDCO) showed activity only against *S. aureus* (Table 1, Appendix A). Three control compounds H(ACO), H(ECO), and H(MCO) were tested and showed no antimicrobial effect. The results showed that the cyanoximates alone have no antimicrobial impact, and the incorporation of the SbPh_4_ fragment into selected backbones enables the antimicrobial potential. Comparison of two (twofold and fourfold) dilutions for each compound reflected concentration-dependent growth inhibition for most of the compounds, except for SbPh_4_(ACO) (*E. coli*), SbPh_4_(ECO) (*P. aeruginosa*), and SbPh_4_(TDCO) (*S. aureus*) (Appendix A). This may reflect a time-dependent antimicrobial affect [43]. According to the inhibition profile, SbPh_4_(ACO), SbPh_4_(ECO), and SbPh_4_(MCO) do not discriminate between Gram-negative and Gram-positive bacteria, suggesting that their broad impact on bacterial cells is independent of the presence or absence of the outer membrane of Gram-negative cells and peptidoglycan of Gram-positive cells. The SbPh_4_(TDCO) activity only against Gram-positive *S. aureus* suggests that the outer membrane of Gram-negative *E. coli* and *P. aeruginosa* presents a sufficient protective barrier for this compound. These results confirm that the new antimony compounds have strong potential as both broad- and narrow-spectrum antimicrobials and provide insights for further synthetic modifications of the compounds to increase their activities.

**Antifungal potential of new tetraphenyl-antimony(V) cyanoximates.** To determine the antifungal potential of the novel antimony compounds against fungal pathogens, we selected two human fungal pathogens—*Cryptococcus neoformans* and *Candida albicans*. Both *C. neoformans* and *C. albicans* are becoming increasingly resistant to currently available antifungals [13,44,45,46], and resistance is being tracked through the CDC’s Emerging Infections Program (EIP). Both strains were subjected to a disc-diffusion assay (Table 2). Among the compounds tested, SbPh_4_(MCO) was the only compound to inhibit the growth of both fungal pathogens *C. neoformans* and *C. albicans*. The remaining compounds inhibited the growth of *C. neoformans* but not *C. albicans* (Table 2, Appendix A). These results confirm that the new antimony compounds have a strong potential, especially against the fungal pathogen *C. neoformans*, but they may need further synthetic modifications to be active against *C. albicans*. 

We observed a pronounced synergistic effect of the presence of organoantimony(V) and cyanoxime moieties in these molecular compounds. The results of the conducted interdisciplinary research warranted filing full patent application No. 63/152,490 “Organoantimony(V) cyanoximate compounds and methods of production and use thereof”, filed on 23 February 2022. On 6 September 2022, the US Patent and Trademark Office informed us that the above-referenced PCT application was published as International Publication No. WO2022/182719 A1, with an International Publication Date of 1 September 2022.

## 3. Materials and Methods


**Chemistry.**


**General Considerations.** Cyanoximes selected for current studies were obtained using published procedures from starting substituted acetonitriles [14], and their purity was verified using TLC, NMR spectroscopy, and elemental analyses of C, H, N content using combustion method (Atlantic Microlab, Norcross, GA). Starting compounds with sulfur-containing cyanoximes Tl(TCO) and Tl(TDCO) were also prepared as we described earlier [34,47]. The source of organoantimony, ~95% purity, was Sb(C6H5)4Br purchased from Aldrich. All other chemicals and organic solvents—*i*-PrOH, CH_3_CN, ether, NaNO_2_, H_2_SO_4_, and HCl from Fisher Scientific and sodium metal from Fluka—were of sufficient quality and used without additional purification. Melting points and thermal decomposition profiles were obtained using thermal analysis method presented below. Electrical conductivity of 0.001 M solutions of SbPh_4_L (L = cyanoximes shown in Figure 2) was measured in CH_3_CN and DMSO at 296 K with aid of the YSI 3100 conductivity meter with Pt-electrode using N(CH_3_)_4_Br and P(C_6_H_5_)_4_Cl salts as calibrants.

**Chemical Synthesis.** Preparation of cyanoximes shown in Figure 2 was conducted according to the published procedures for HACO [30], HTCO [21], HTDCO [47], HECO [48], HMCO [31], and isomeric pyridyl-cyanoximates [35]. The source of antimony was bromide of tetraphenyl-antimony(V). We determined that the best synthetic route to organometallic cyanoximates is a metathesis reaction of Ph4Br with silver or thallium cyanoximates, as shown in Figure 2. The latter salt was used for the S-containing cyanoximes HTCO and HTDCO for which corresponding silver(I) salts do not exist due to fast decomposition with the formation of Ag_2_S, while Tl(TCO) and Tl(TDCO) [34,47] can be easily obtained and are thermally and light stable. The typical preparation is given only for two tetraphenyl-antimony(V) cyanoximates. 

Silver(I) salts of ligands listed in Figure 2 are light-stable and were obtained in aqueous solutions in two steps as shown in Figure 3A using AgNO_3_ [32,49], while two thallium(I) salts were prepared from Tl_2_CO_3_ (Figure 3B). The actual appearance of dark-yellow and orange TlL (L = TCO, TDCO) can been in Appendix A.

**SbPh_4_(MCO).** The preparation of SbPh_4_(MCO) was conducted by dissolving 0.201 g (0.394 mM) of SbPh_4_Br in 10 mL of dry acetonitrile in a 25 mL Erlenmeyer flask to which 0.120 g (0.413 mM, being in slight excess because of heterogeneous reaction) of finely ground, solid Ag(MCO) [31] was added in small portions under stirring at room temperature. This reaction requires red-light (or dark) conditions to prevent photodecomposition of the light-sensitive byproduct AgBr. Use of an ultrasound bath is necessary to assure proper reagent mixing in this heterogeneous reaction. Following sonication, removal of the white AgBr precipitate was found to be the most convenient by centrifugation using nylon 0.45 micron membrane inserts into 5 mL Eppendorf tubes (Appendix A). Thus, the cloudy reaction mixture of the desired organoanimtony(V) cyanoximate in CH_3_CN solution and fine powder of AgBr was transferred into nylon Eppendorf tubes and centrifuged in a Thermo Scientific Sorvall Legends Micro 17 at 10,000 rpm for 3 min. The colorless and transparent solution containing SbPh_4_(MCO) was carefully pipetted from the tube into 25 mL beaker placed into a charged with a paraffin (for absorption of CH_3_CN) vacuum desiccator for further crystallization. After 1 week, colorless block-type crystals of the complex suitable for the X-ray analysis were harvested from the beaker. 

**SbPh_4_(TCO).** Synthesis of this compound involved 0.250 g (0.490 mM) of SbPh_4_Br and 0.170 g (0.511 mM, slight excess due to heterogeneous reaction condition) of a yellow powder of Tl(TCO) mixed together with 10 mL of dry CH_3_CN in a 25 mL beaker at room temperature. The preparation of Tl(TCO) is described in Appendix A, while Tl(TDCO) synthesis was carried out according to a published procedure [30] with results of the elemental analysis displayed in Appendix A. Since Tl(I) salts are non-oxidizers and not light-sensitive, the reaction was carried out conveniently on a bench top. Thorough mixing of reagents in this heterogeneous reaction was performed in an ultrasound bath within 10 min. The content of the beaker as a fine yellow suspension was placed in Eppendorf tubes and centrifuged within 3 min at 10,000 rpm, leading to clean separation of the white TlBr and transparent yellow solution of target compound. Careful pipetting of this solution into another beaker followed by its concentration in a vacuum desiccator, charged with paraffin to absorb CH_3_CN, afforded light-yellow transparent plates (Appendix A). 

Other compounds were prepared in a similar fashion. Information regarding the synthesized compounds’ color, yield, and results of elemental analyses presented in Table 3.

**Safety** **Note:**
*Although we did not encounter any problems during many years of laboratory work and handling, special care should be taken during work with thallium compounds because of their high toxicity [50,51,52]. Both Tl(I) carbonate and cyanoximates are water-soluble compounds, which emphasizes the absolute necessity of always wearing protective gloves when working with these compounds. However, no respiratory tract covers are required since Tl(I) compounds are ionic and not volatile.*



**Spectroscopic Studies.**


***Vibrational spectra***. The IR spectra of all the prepared compounds reported herein were recorded at ambient laboratory temperature using small quantities (5–10 mg) of solid samples of studies compounds with the help of a Bruker ATR FT spectrophotometer. The resolution was set to 4 cm^−1^ with 64 scans in the range of 400–4000 cm^−1^. An atmospheric compensation and baseline correction were applied for data treatment. Raman spectra were recorded on the Bruker R70 FT-IR/Raman complex at room temperature. The power of the laser and the number of repetitions were varied depending on the sample in the ranges of 100–250 mW and 64–1024 scans, respectively. The combined results of vibrational spectroscopy as assigned peaks of the most important vibrations in spectra are presented in Appendix A.

***UV/visible spectra***. The electronic spectra of the compounds were recorded in solutions using the HP 8354 diode array UV/visible spectrophotometer operating in the range of 200–1100 nm, typically at room temperature. In cases where variable-temperature spectra were needed, a high-precision Peltier Quantum Northwest thermocontroller was used with 1 cm quartz cuvettes. 

***NMR spectra***. The ^1^H- and ^13^C{^1^H}-NMR spectra were obtained on a Varian Inova-400 MHz spectrometer at room temperature in CD_2_Cl_2_ containing TMS as an internal reference. Data are tabulated in Appendix A, whilst several selected spectra are displayed in Appendix A. 


**X-ray crystallography.**


Crystal structures were determined for all the eight organoantimony(V) compounds. Suitable single crystals of organoantimony(V) cyanoximates were grown at 4 °C exclusively using the vapor diffusion method. Briefly, acetonitrile solutions of compounds were placed into the inner tube, and vapors of anhydrous ether were used as the crystallizing agent from the outer tube with a screw cap. Normally within 2–3 weeks after setting up, transparent well-shaped crystals suitable for crystallographic studies appeared on the walls in the inner tube. All crystals suitable for studies were placed on plastic MiTeGen holders attached to the copper pin on the goniometer head of the Bruker APEX-2 diffractometer, equipped with a SMART CCD area detector. The intensity data were collected at low temperature. Data collection was performed in ω scan mode using the Mo tube (K_α_ radiation; λ = 0.71073 Å) with a highly oriented graphite monochromator. Intensities were integrated from four series of 364 exposures, each covering 0.5° steps in ω at 20–60 s of exposure time depending on the crystal diffracting power, with the total dataset being a sphere. The space group determination was conducted with the aid of the XPREP software. The absorption correction was performed in a few cases by a crystal face indexing procedure with the help of a video microscope (Appendix A), followed by numerical input into the SADABS program. Ultimately, in the case of a small crystal specimen and their irregular shape, we used the multi-scan method. Both cited programs are included in the Bruker AXS software package [53]. All structures were solved using direct methods and refined by least squares on weighted *F*^2^ values for all reflections using the SHELXTL program. In several structures, due to the high crystal quality, it was possible to identify all H-atoms on the electron density map. However, in some structures, H-atoms were placed in calculated positions in accordance with the hybridization state of a hosting carbon atom and were refined isotropically. No apparent problems or complications were encountered during the structure solutions and refinement as evident from very positive PLATON checkCIF reports. The crystal data for six structures of Sb-cyanoximates are presented in Table 4, while their ASUs are shown in Figure 3. Values of selected bond lengths and valence angles are presented in the respective figures in Appendix A. All drawings of the crystal structures were completed using the ORTEP 3v2 [54] and freely available CCDC Mercury software packages. All determined crystal structures were deposited at CCDC (England) under the following numbers: SbPh_4_(MCO)—2012156 (as refcode SURYIQ); SbPh_4_(ECO)—2012153 (as SURXUB), SbPh_4_(4PCO)—2011863 (as SUPPAX), SbPh_4_(3PCO)—2011860 (as SUPNID), SbPh_4_(2PCO)—2011861 (as SUPNOJ), and SbPh_4_(ACO) (as ZASVUK02). The PLATON checkCIF reports are presented at the end of the Appendix A.

**Thermal Analysis**. Thermal stability of the obtained metal complexes was assessed using a Q-600 TG/DSC analyzer (TA Instruments) under N_2_ flow of 100 mL/min at a 10 °C/min heating rate. Heating of samples was carried out to 800 °C, at which complete decomposition of the samples was attained. The final product of decomposition is elemental antimony, which appears as a spongy mass inside the crucible.


**Biological studies.**


Susceptibility testing for bacteria and fungi using disc-diffusion assay. Discs to be tested were prepared by applying the synthesized compounds dissolved in acetonitrile at a concentration of 0.05 µM onto 100% cotton paper discs of 7 mm width that were pre-dried in a desiccator over H_2_SO_4_. The loaded discs were then placed on a Teflon block to allow them to dry completely. Five Sb compounds, SbPh_4_(ACO), SbPh_4_(ECO), SbPh_4_(MCO), SbPh_4_(TDCO), and SbPh_4_(TCO), and three control compounds, H(ACO), H(ECO), and H(MCO), were included. Dried discs with no material loaded were used as negative controls.

**Bacterial strains and growth media.** Low-salt Luria–Bertani (LB) medium was used for antimicrobial susceptibility testing. The bacterial stains included Gram-negative *Escherichia coli* strain S17(O113:H4) isolated from a chick liver with septicemia and lab-adapted burn-wound isolate *Pseudomonas aeruginosa* strain PAO1, and Gram-positive methicillin resistant *Staphylococcus aureus* strain NRS70. All strains were maintained in 10% skim milk at −80 °C. Before each experiment, the bacterial strains were inoculated onto LB agar from frozen stocks and grown overnight at 37 °C, from which isolated colonies were picked for inoculating LB broth and the subsequent testing.

**Susceptibility testing for bacteria.** Discs to be tested were prepared as described above. Bacterial cultures were grown in LB broth shaking at 200 rpm for 12 h at 37 °C. The optical density of the cultures at 600 nm (OD_600 nm_) was measured and adjusted to 0.1 in LB broth, and 100 µL of the cultures were spread on LB plates. The plates were allowed to dry for 10 min before impregnating filter paper discs containing test compounds using sterile forceps. As a control, 10 µL of phosphate-buffered saline (PBS) was applied onto the discs. The plates were incubated at 37 °C for 24 h. Antibacterial activity was determined by measuring the clearance zones surrounding the discs. Considering irregular shapes of some clearances, three measurements for each sample were performed and averaged. Each experiment was conducted using three biological replicates, and the clearance zones were reported as the mean + standard error of the mean (SEM). 

**Fungal strains and growth media.***Cryptococcus neoformans* strain H99 (serotype A, mating type α) and *Candida albicans* strain SC5314 were recovered from 15% glycerol stocks stored at −80 °C and were cultured on yeast extract–peptone–dextrose (YPD) plates (BD Difco; Franklin Lakes, NJ, USA).

**Susceptibility testing for fungi.** Discs to be tested were prepared as described above. Fungal cultures were taken from YPD plates and grown in YPD broth shaking at 180 rpm for 18 h at 30 °C. Cells were centrifuged and washed 3× with sterile PBS. Cells were quantified on a hemacytometer using trypan blue dye exclusion, and cultures were adjusted to 1 × 10^6^ cells/mL in PBS. A volume of 100 µL of the culture was spread onto each YPD plate. The plates were allowed to dry for 10 min before placing filter paper discs containing test compounds using sterile forceps. A volume of 10 µL of PBS was applied to each disc to aid in diffusion. The plates were incubated at 30 °C for 48 h. Antifungal activity was determined by measuring the clearance zones surrounding the discs. Three measurements for each sample were performed and averaged, and each experiment was conducted using three biological replicates. Clearance zones are reported as the mean ± SEM.

## 4. Conclusions

For the first time, a series of novel organoantimony(V) cyanoximates with specially selected features was obtained and thoroughly studied in this interdisciplinary research aimed at a search for new non-antibiotic antimicrobials. As a summary of the conducted investigations, we report the following:(1)We designed high-yield synthesis of a series of eight novel tetraphenyl-antimony(V) cyanoximates of SbPh_4_(L) composition using an effective and clean heterogeneous metathesis reaction involving solid silver(I) or thallium(I) compounds in acetonitrile solutions.(2)The prepared complexes represent crystalline white molecular solids (two compounds with thioamide-cyanoximes TCO^−^ and TDCO^−^ are yellow) well soluble in organic solvents.(3)All the synthesized compounds were characterized by ^1^H- and ^13^C-NMR, vibrational spectroscopy (IR and Raman), variable-temperature UV/visible spectroscopy, thermal analysis, and X-ray analysis.(4)Several correlations between pK_a_ values of starting cyanoxime and solid-state structures of SbPh_4_(L) were determined (a) between off-plane Sb–O distance in trigonal bipyramids of compounds and (b) between Sb–O distances.(5)The crystal structures were determined for all eight new tetraphenyl-antimony(V) cyanoximates and show monodentate O-binding of the anion to the central atom.(6)All synthesized compounds were thermally stable in the solid state and in solutions up to 140 °C, which is an important property in light of their thermal sterilization for intended applications as topical antimicrobial agents added to coating and paints.(7)For the first time, a series of organometallic Sb-compounds of SbPh_4_L composition were carefully designed for in vitro antimicrobial testing.(8)When tested for antibacterial potential, SbPh_4_(ACO) and SbPh_4_(ECO) showed activity against three selected bacterial pathogens, *E. coli*, *P. aeruginosa*, and *S. aureus*. known for their outstanding antibiotic resistance. SbPh_4_(ACO), SbPh_4_(ECO), and SbPh_4_MCO showed a broad impact against the three bacteria, and SbPh_4_(TDCO) was active only against Gram-positive *S. aureus*.(9)Antifungal studies showed that SbPh_4_(MCO) was the only compound to inhibit the growth of both fungal pathogens *C. neoformans* and *C. albicans*, also known for their increasing resistance to antimicrobials. The other compounds inhibited the growth of *C. neoformans* but not *C. albicans*.

Table of content picture (General formula for the first series of new non-antibiotic antimicrobial compounds.):



Synopsis:

For the first time, a series of novel organoantimony(V) cyanoximates with specially selected features was obtained and thoroughly characterized using spectroscopic methods, as well as thermal and X-ray analyses. Compounds were found to be stable in the solid state and in solutions, representing new non-antibiotic compounds active against both pathogenic microorganisms and fungi. 

## Data Availability

All results of this study and data from the Appendix A are available to the public via the journal’s portal.

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
