# Peer review of "Non-Antibiotic Antimony-Based Antimicrobials"

_molecules, 2022, doi:10.3390/molecules27217171_

Round 1
Reviewer 1 Report
The experimental work presented in this manuscript are of good quality and the work is significant and should be of interest to the readership. The aspects of this manuscript that must be addressed prior to publication are mostly stylistic but are important.
The authors should use appropriate and current terminology: there is no reason to call "coordination complexes" "Werner-type complexes", particularly given that the antimony compounds presented are best described as probably covalent, organometallic compounds (as the authors note!). The distinction between the two types of bonding presented in Figure 4 is unnecessary in a journal such as "Molecules" and should be removed.
Similarly, the pnicotogen compounds are best described as being in "group 15" rather than in the anachronistic "group V".
Many of the chemical drawings are very pixelated and should be redrawn at higher resolution.
Although it is readable and understandable, some of the sentence construction used in this manuscript is less than ideal and can be improved.
Author Response
Reply on reviewer #1 comments on our manuscript 1950607 for “Molecules”:
A short clarification of differences between Werner-type complexes and organometallic compounds [that is what we had obtained!] was added to the paper. We respectfully disagree with this reviewer and provided some additional, and hopefully convincing, writing. It is essential for the explanation of compounds’ biological activity! Numerous previously studied Sb(V) compounds with carbohydrates and other molecules did not have biological activity namely die to hydrolysis reactions. Organometallic compounds have great advantage in such situation since Sb-C bond is very stable. Thus, we need to keep this “old fashioned” classification and description in place in the manuscript because observed antimicrobial activity is associated just with the fact of stability in hydrolytic environment of Sb-cyanoximates.
Group V mentioning has replaced with the Group XV as was suggested.
Pixelated Figure 3 of complain was re-drawn to attain readability.
Reviewer 2 Report
A very good paper, with a very interesting and (potentially) useful topic. As far as I can check this, the investigations have been performed properly and meticulously. The English language is very good. Even people who do not know English perfectly can understand also very technical parts of the manuscript without difficulties. Thus, the manuscript could be published without changes, but ....., what about the Author Contributions chapter.
It is very strange. Who exactly are XX, YY, and ZZ? ;-) These individuals should be identified. The chapter should be extensively rewritten, contributions of all Authors must be clearly indicated. At this stage only contributions from XX, YY, and ZZ are presented ;-))), and the work has not three but seven Authors! Please remember "Authorship must be limited to those who have contributed substantially to the work reported."
Author Response
Reply on reviewer #2 comments on our manuscript 1950607 for “Molecules”:
This reviewer made a comment (quote): It is very strange. Who exactly are XX, YY, and ZZ? ;-) These individuals should be identified. The chapter should be extensively rewritten, contributions of all Authors must be clearly indicated. At this stage only contributions from XX, YY, and ZZ are presented ;-))), and the work has not three but seven Authors! Please remember "Authorship must be limited to those who have contributed substantially to the work reported."
It is important to note that this paper reflects results of an interdisciplinary work between two institutions in two different states of the USA, and started in 2017. Clearly this work involved chemists and microbiologists that were experts in antimicrobial and antifungal studies. Based on a suggestion from this reviewer we made additional section at the end of the paper:
Author Contributions: Kevin Pinks prepared all chemically characterized all studied compounds, Nikolay Gerasimchuk initiated the project and provided its overall supervision, Tarosha Salpadoru and Olga Michka performed antibacterial studies, Marianna Patrauchan directed antibacterial studies, edited the manuscript, Kaitlyn Cotton performed antifungal studies, Karen Wozniak directed antifungal studies, edited the manuscript.
We hope that now that question about involved people, their number and affiliations with carried out responsibilities is answered.